# SONIC: SPECTRAL ORIENTED NEURAL INVARIANT CONVOLUTIONS

**Gijs Joppe Moens**[1,2]**, Regina Beets-Tan**[1,3]**, Eduardo H. P. Pooch**[1,3*]
[1] Netherlands Cancer Institute, [2] University of Amsterdam, [3] Maastricht University
{g.moens, r.beetstan, e.pais.pooch}@nki.nl

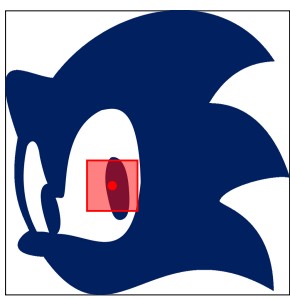 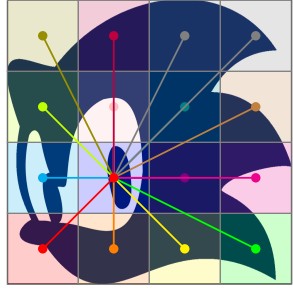 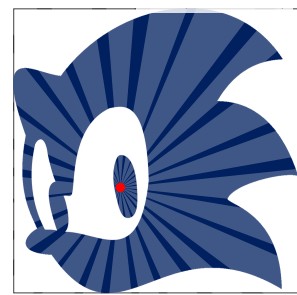

**Local convolution**    **Self-attention**    **SONIC (Ours)**

## ABSTRACT

Convolutional Neural Networks (CNNs) rely on fixed-size kernels scanning local patches, which limits their ability to capture global context or long-range dependencies without very deep architectures. Vision Transformers (ViTs), in turn, provide global connectivity but lack spatial inductive bias, depend on explicit positional encodings, and remain tied to the initial patch size. Bridging these limitations requires a representation that is both structured and global. We introduce **SONIC (Spectral Oriented Neural Invariant Convolutions)**, a continuous spectral parameterisation that models convolutional operators using a small set of shared, orientation-selective components. These components define smooth responses across the full frequency domain, yielding global receptive fields and filters that adapt naturally across resolutions. Across synthetic benchmarks, large-scale image classification, and 3D medical datasets, SONIC shows improved robustness to geometric transformations, noise, and resolution shifts, and matches or exceeds convolutional, attention-based, and prior spectral architectures with an order of magnitude fewer parameters. These results demonstrate that continuous, orientation-aware spectral parameterisations provide a principled and scalable alternative to conventional spatial and spectral operators.

## 1 INTRODUCTION

Human visual processing effortlessly recognises objects, interprets motion, and comprehends complex scenes across varying orientations, scales, resolutions, and degraded conditions—capabilities where computer vision methods still fall short. Bridging this gap remains a central challenge, driving the development of models that more closely approximate the versatility and robustness of human perception.

Convolutional Neural Networks (CNNs) (LeCun et al., 2015) rely on fixed-size kernels scanning local patches. While effective for capturing local features, this design limits long-range dependency modelling without very deep architectures (Luo et al., 2017) and is sensitive to geometric variations such as translations, rescalings, rotations, and distortions (Azulay & Weiss, 2018). Vision Transformers (ViTs) (Dosovitskiy et al., 2021) overcome locality constraints via self-attention over image

---

*Code available at https://github.com/GijsMoens/Sonic

patches, but their quadratic cost in image area poses challenges for high-resolution inputs. Moreover, ViTs lack spatial inductive biases, requiring explicit positional encodings, and their accuracy–compute trade-off is tied to the chosen patch size. With the proposed method, which enables global receptive fields using significantly fewer parameters, we aim to narrow this gap and move toward resolution-invariant perception, drawing inspiration from the robustness of human visual processing.

**Contribution.** In this paper, we introduce a theoretically grounded spectral framework for multi-dimensional signals that naturally provides global receptive fields, full convolutional expressiveness, and inherent resolution invariance, offering a lightweight yet versatile foundation that can support progress toward more scalable and adaptable vision models. The remainder of this paper is organised as follows. Section 2 introduces the mathematical preliminaries and related work. Section 3 presents the formulation of the SONIC approach together with implementation details. Section 4 reports the experimental results. Section 5 discusses the limitations of the proposed method and outlines directions for future research.

## 2 BACKGROUND

Modern vision requires integrating information over long spatial ranges, yet standard convolutions remain bounded by local receptive fields. This section reviews operators that achieve global receptive fields and develops the spectral framework underpinning our approach.

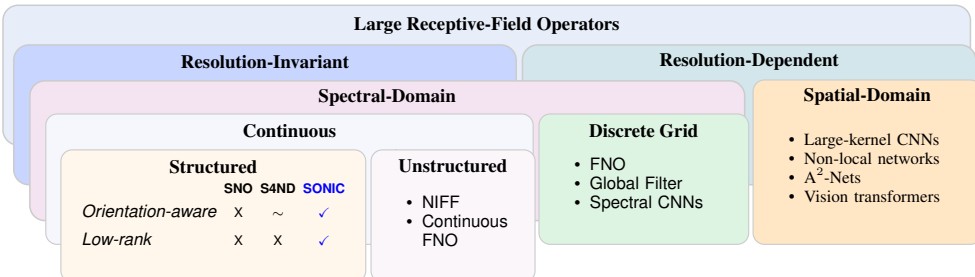

Figure 1: Taxonomy of large receptive-field operators.

**Spatial-domain operators.** In the spatial domain, enlarging the receptive field requires expanding the discrete kernel support. Large-kernel convolutions (Ding et al., 2022), dilated convolutions (Yu & Koltun, 2015), and non-local methods (Wang et al., 2018) each target long-range interactions, but all operate on a fixed sampling grid whose locality assumptions break down over large spatial ranges. Self-attention (Chen et al., 2018; Dosovitskiy et al., 2021) computes pairwise interactions across all positions, yet scales quadratically with the number of tokens. As resolution increases, both large kernels ($O(K^2)$) and attention become prohibitively costly in high-resolution domains.

**Spectral-domain operators.** An alternative paradigm achieves global receptive fields by representing operators directly in the frequency domain (Rippel et al., 2015). This approach exploits the fact that every linear, shift-invariant operator on $\mathbb{R}^D$ is fully characterised by its frequency response, enabling information to propagate globally through a single frequency-wise multiplication.

Let $D \in \mathbb{N}$ and consider vector-valued signals

$$x : \mathbb{R}^D \to \mathbb{C}^{C_{\text{in}}}, \qquad y : \mathbb{R}^D \to \mathbb{C}^{C_{\text{out}}}.$$

For a sufficiently regular scalar function $f : \mathbb{R}^D \to \mathbb{C}$, the Fourier transform is

$$\mathcal{F}_D[f](\boldsymbol{\omega}) = \int_{\mathbb{R}^D} f(\mathbf{x}) \, e^{-i\,\boldsymbol{\omega}\cdot\mathbf{x}} \, d\mathbf{x}, \qquad \boldsymbol{\omega} \in \mathbb{R}^D, \tag{1}$$

and extends component-wise to vector-valued functions. A linear, shift-invariant operator acting on $x$ has a convolution representation:

$$y(\mathbf{x}) = \int_{\mathbb{R}^D} k(\mathbf{x} - \mathbf{z}) \, x(\mathbf{z}) \, d\mathbf{z}, \tag{2}$$

where $k : \mathbb{R}^D \to \mathbb{C}^{C_{\text{out}} \times C_{\text{in}}}$. The Convolution Theorem gives

$$\mathcal{F}_D[y](\boldsymbol{\omega}) = \widehat{k}(\boldsymbol{\omega}) \, \mathcal{F}_D[x](\boldsymbol{\omega}), \qquad \boldsymbol{\omega} \in \mathbb{R}^D, \tag{3}$$

where $\widehat{k}(\boldsymbol{\omega})$ is the Fourier transform of a spatial kernel in the classical convolution setting. In spectral neural methods, however, we do not constrain the operator to arise from any finite-support spatial kernel. Instead, we define the spectral kernel directly by

$$\widehat{k}(\boldsymbol{\omega}) := \widehat{H}(\boldsymbol{\omega}), \tag{4}$$

where $\widehat{H}(\boldsymbol{\omega})$ is the learnable frequency response of the operator. This viewpoint treats $\widehat{H}$ as the primary parametrisation, enabling general global and resolution-invariant operators beyond those that correspond to discrete spatial kernels. In practice, images are sampled on a discrete grid. Let the spatial domain be discretised using $N_1, \ldots, N_D$ samples along each axis, with pixel spacings $\Delta_1, \ldots, \Delta_D$. The corresponding DFT frequency sets are given by

$$\Omega_d = 2\pi \left\{ \frac{k_d}{N_d \, \Delta_d} \ \middle| \ k_d = -\left\lfloor \tfrac{N_d}{2} \right\rfloor, \ldots, \left\lceil \tfrac{N_d}{2} \right\rceil - 1 \right\}, \quad d = 1, \ldots, D. \tag{5}$$

The full frequency grid is the Cartesian product $\Omega = \Omega_1 \times \cdots \times \Omega_D$, containing $N = N_1 \cdots N_D$ discrete frequencies. The DFT samples $\widehat{x}$ are defined at frequencies $\boldsymbol{\omega}_n \in \Omega$.

**Resolution invariance.** We formalise resolution invariance by defining the operator via a continuous spectral symbol that is independent of the sampling grid. Let

$$\widehat{H}_\theta : \mathbb{R}^D \to \mathbb{C}^{C_{\text{out}} \times C_{\text{in}}} \tag{6}$$

be a continuous function parameterised by $\theta$. Given a discretisation $(N, \Delta)$ with Fourier grid $\Omega_{N,\Delta}$, the discretised operator is obtained via sampling:

$$\widehat{y}^{(N,\Delta)}(\boldsymbol{\omega}_n) = \widehat{H}_\theta(\boldsymbol{\omega}_n) \, \widehat{x}^{(N,\Delta)}(\boldsymbol{\omega}_n), \qquad \boldsymbol{\omega}_n \in \Omega_{N,\Delta}. \tag{7}$$

We term the operator resolution-invariant if $\theta$ depends only on the underlying physics of the layer, not on the discretisation $(N, \Delta)$. Changing resolution then simply corresponds to resampling the same continuous function $\widehat{H}_\theta$ on a new grid. GFNet (Rao et al., 2021) and FNO (Li et al., 2021) parameterise $\widehat{H}$ directly on the discrete FFT grid: GFNet learns a complex mask of size $N$, and FNO learns a fixed number of low-frequency coefficients. Since these coefficients correspond to specific frequency indices, changing resolution alters the operator itself. Thus, such models do not define a true resolution-invariant convolution operator.

**Continuous spectral operators.** A principled alternative defines the operator in the continuous Fourier domain and evaluates it on whatever discrete grid the data provides, yielding a truly resolution-invariant parameterisation. Two families appear in the literature:

- **Unstructured continuous operators** learn $\widehat{H}(\boldsymbol{\omega})$ as a general continuous function, e.g. via an MLP (Grabinski et al., 2024; Kabri et al., 2023). These are maximally flexible but typically isotropic and parameter-inefficient, as they offer little inductive bias for orientation or cross-channel structure.

- **Structured continuous operators** impose additional structure through basis expansions or separability assumptions. Examples include SNO (Fanaskov & Oseledets, 2024) and S4ND (Nguyen et al., 2022), whose axis-aligned constructions couple nearby frequencies and improve parameter efficiency. However, their separability limits the capture of oriented or anisotropic patterns along general directions in frequency space. We provide the frequency-domain form of S4ND in Appendix C.2.

Since natural images contain oriented structures (edges, textures, oscillations) that correspond to directional features in frequency space, axis-aligned or tensor-product constructions are insufficient. We therefore introduce SONIC: Spectral Oriented Neural Invariant Convolutions, a Structured Continuous Spectral Operator that parameterises $\widehat{H}_\theta(\boldsymbol{\omega})$ as a superposition of directional modes, yielding resolution-invariant, parameter-efficient filters inherently adapted to the anisotropic structure of natural signals.

## 3 METHOD

**Overview.** Starting from linear time-invariant (LTI) systems, we derive a compact spectral representation for $D$-dimensional signals that models shift-invariant operators through a shared low-rank structure: oriented spectral transfer functions are applied at each frequency and mixed across channels by learned matrices $B$ and $C$.

### 3.1 FORMULATION

To make the connection to state-space models precise, consider the continuous-time LTI system:

$$\dot{\mathbf{x}}(t) = \mathbf{A}\mathbf{x}(t) + \mathbf{B}\mathbf{u}(t), \qquad \mathbf{y}(t) = \mathbf{C}\mathbf{x}(t), \tag{8}$$

with zero initial condition. Its impulse response is obtained by setting $\mathbf{u}(t) = \delta(t)$:

$$\mathbf{K}(t) = \mathbf{C}\,e^{\mathbf{A}t}\,\mathbf{B}, \qquad t \geq 0. \tag{9}$$

The output equals the convolution of the input with the impulse response:

$$(\mathbf{K} * \mathbf{u})(t) = \int_0^{\infty} \mathbf{C}\,e^{\mathbf{A}\tau}\,\mathbf{B}\,\mathbf{u}(t - \tau)\,d\tau. \tag{10}$$

Taking the Laplace transform of the impulse response (derivations provided in Appendix C),

$$H(s) = \mathcal{L}\{\mathbf{K}(t)\}(s) = \mathbf{C}(s\mathbf{I} - \mathbf{A})^{-1}\mathbf{B}. \tag{11}$$

We use this resolvent form as a modelling template: replacing the scalar Laplace variable $s$ with the $D$-dimensional spatial frequency $\boldsymbol{\omega}$ yields an analytic spectral parameterisation that inherits the smooth behaviour of resolvent filters.

Let $x \in \mathbb{R}^{C \times N_1 \times \cdots \times N_D}$ and $y \in \mathbb{R}^{K \times N_1 \times \cdots \times N_D}$, with DFT $\widehat{x} = \mathcal{F}_D[x]$ and $y = \mathcal{F}_D^{-1}[\widehat{y}]$. Central to our method is the transfer function $T_m(\boldsymbol{\omega})$, which defines the frequency response of a single mode. For each mode $m = 1, \ldots, M$ we set

$$T_m(\boldsymbol{\omega}) = \frac{1}{i\,s_m\,(\boldsymbol{\omega} \cdot \boldsymbol{v}_m) - a_m + \tau_m\,\|(I - \boldsymbol{v}_m \boldsymbol{v}_m^{\top})\boldsymbol{\omega}\|_2^2}, \tag{12}$$

where each mode is parameterised by: the unit orientation $\boldsymbol{v}_m \in \mathbb{R}^D$; scale $s_m > 0$ (spectral selectivity); damping $\mathrm{Re}(a_m) < 0$ and oscillation $\mathrm{Im}(a_m)$; and transverse penalty $\tau_m \geq 0$ (off-axis decay). The denominator replicates the resolvent structure of an LTI system, substituting the Laplace variable with the oriented frequency component $i\,s_m(\boldsymbol{\omega} \cdot \boldsymbol{v}_m)$ and adding a transverse decay for anisotropic filtering. Each mode additionally carries a learnable Butterworth bandwidth $b_m$ to suppress aliasing near the Nyquist limit (Appendix A).

Rather than learning an unconstrained response $\widehat{H}(\boldsymbol{\omega})$ for every frequency, SONIC factorises the spectral operator through $M$ shared modes with entrywise form:

$$\widehat{H}_{k,c}(\boldsymbol{\omega}) = \sum_{m=1}^{M} C_{km}\,T_m(\boldsymbol{\omega})\,B_{mc}. \tag{13}$$

where $B \in \mathbb{C}^{M \times C}$ and $C \in \mathbb{C}^{K \times M}$. Given this factorised spectral response, the frequency-wise filtering applied to the input DFT is

$$\widehat{y}_k(\boldsymbol{\omega}) = \sum_{c=1}^{C} \widehat{H}_{k,c}(\boldsymbol{\omega})\,\widehat{x}_c(\boldsymbol{\omega}), \qquad k = 1, \ldots, K, \ \boldsymbol{\omega} \in \Omega, \tag{14}$$

where $\widehat{H}_{k,c}(\boldsymbol{\omega})$ is the frequency response of the $(c \to k)$ channel filter. This low-rank decomposition enables expressive yet parameter-efficient filtering. The spatial output is added to a learnable skip projection and passed through a pointwise nonlinearity:

$$x^{(\ell+1)} = \sigma\big(y^{(\ell)} + W_s x^{(\ell)}\big). \tag{15}$$

Multiple SONIC blocks can be stacked for depth-wise expressivity. Although SONIC is not a state-space model, restricting its orientations to coordinate axes recovers the multidimensional SSM form (Appendix C.2).

## 3.2 INTUITION

Each SONIC mode is a compact, analytic spectral filter with a few interpretable parameters (illustrated in Figure **??**). A unit vector $\boldsymbol{v}_m$ sets the mode's preferred direction; any $\boldsymbol{\omega}$ decomposes into along- and across-orientation components:

$$\omega_{\|m} := \boldsymbol{\omega} \cdot \boldsymbol{v}_m, \qquad \boldsymbol{\omega}_{\perp m} := (I - \boldsymbol{v}_m \boldsymbol{v}_m^\top)\boldsymbol{\omega}.$$

The mode emphasises slowly varying content along $v_m$ and suppresses energy across it, producing a spatially elongated kernel sensitive to oriented structure. The scale $s_m$ controls spectral selectivity: small values give a broad, smoothing response while large values narrow the passband to fine along-axis detail. The complex coefficient $a_m$ governs dynamics—its real part controls damping (stability) and its imaginary part introduces oscillatory structure, letting the mode capture repetitive patterns. The transverse penalty $\tau_m \geq 0$ suppresses off-axis leakage, sharpening directional selectivity. The modes form a shared dictionary of directional behaviours; mixing matrices $\mathbf{B}$ and $\mathbf{C}$ let each channel pair draw a unique superposition, inducing parameter sharing across both frequencies and channels.

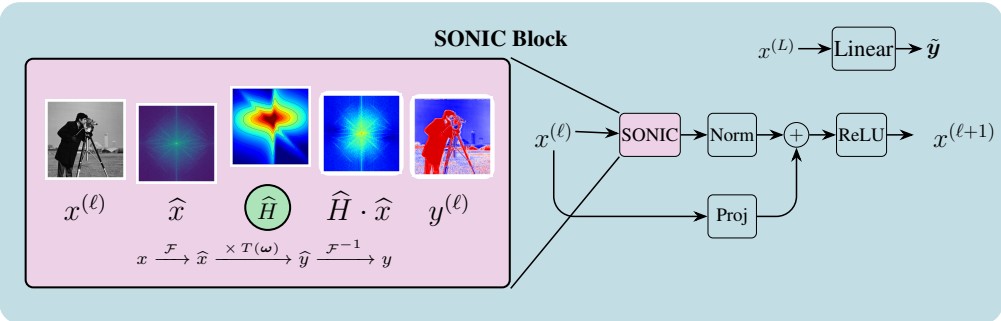

(a) A SONIC block: FFT, spectral modulation by $\widehat{H}$, inverse FFT, normalization, and residual fusion.

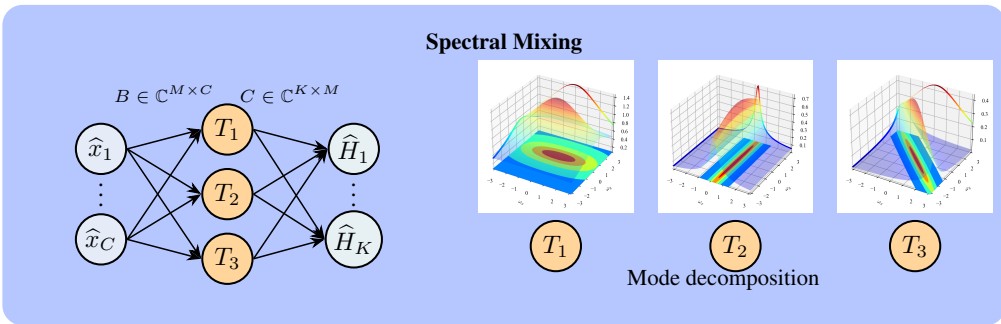

(b) Low-rank spectral mixing: $\widehat{H}_{k,c}(\boldsymbol{\omega}) = \sum_m C_{km} T_m(\boldsymbol{\omega}) B_{mc}$, with $B$ and $C$ mixing channels into and out of $M$ shared modes.

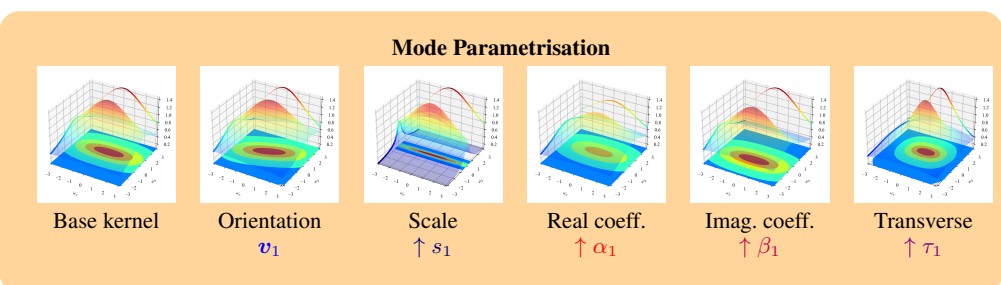

(c) Each mode $T_m$ is controlled by orientation, scale, complex coefficients, and transverse decay. Shown: parameter sweep for $T_1$.

Figure 2: SONIC overview: (a) Residual Block, (b) Spectral Mixing, and (c) Learnable Spectral Modes.

**Resolution Invariance**    Crucially, all of these filters are parameterized directly in the continuous spectral domain. This means their definition does not depend on the size or sampling rate of the image: defining filters as continuous functions of $\omega$ decouples them from any particular grid size or sampling rate; the same response formula is evaluated on whatever DFT grid the data induces, yielding a resolution-invariant filter. This distinguishes our approach from spatial-domain kernels, whose size and shape are tied to a fixed grid. We made some minor adjustments to ensure resolution invariance: To make the directional parameters resolution invariant, we express directions in physical units and normalize:

$$D_\Delta \;=\; \mathrm{diag}(\Delta_1, \ldots, \Delta_D), \qquad \tilde{\boldsymbol{v}}_{\boldsymbol{m}} \;=\; D_\Delta^{-1}\boldsymbol{v}_{\boldsymbol{m}}, \qquad \hat{v}_m \;=\; \frac{\tilde{\boldsymbol{v}}_{\boldsymbol{m}}}{\|\tilde{\boldsymbol{v}}_{\boldsymbol{m}}\|_2}. \tag{16}$$

This formulation provides flexibility with respect to resolution, which can also be exploited during training as proposed in Nguyen et al. (2022). Beyond training efficiency, resolution-aware parameterisation is particularly relevant in domains where resolution dependence is intrinsic, such as medical imaging, remote sensing, and microscopy.

**Computation.**    The number of learnable real scalars is:

$$\underbrace{2KM}_{C^{\mathrm{re}}, C^{\mathrm{im}}} + \underbrace{2MC}_{B^{\mathrm{re}}, B^{\mathrm{im}}} + \underbrace{(4+D)M+1}_{a^{\mathrm{re}}, a^{\mathrm{im}}, s, v, \tau \,\in\mathbb{R}^D} .$$

Crucially, this count scales with $M(C+K)$ and is *independent of spatial resolution*. In contrast, methods that parameterise filters on the discrete FFT grid (e.g. GFNet) require $\mathcal{O}(N_1 \times \cdots \times N_D)$ parameters per layer, which grows rapidly with resolution and dimensionality. This distinction explains why SONIC's parameter advantage is modest on low-resolution 2D benchmarks but becomes pronounced in 3D volumetric settings.

The FFT has per-transform cost $\mathcal{O}(N \log N)$ for a single (complex) channel. The spectral forward pass performs one DFT per input channel and one inverse DFT per output channel, plus $\mathcal{O}(M(C+K))$ complex multiplications per frequency. The forward pass consists of one DFT per input channel and one inverse DFT per output channel, with cost

$$\mathcal{O}(CN \log N) \quad \text{and} \quad \mathcal{O}(KN \log N),$$

where $N = \prod_{d=1}^{D} N_d$ is the total number of spatial points. In addition, frequency-wise multiplications incur a cost of

$$\mathcal{O}\big(M(C+K)N\big),$$

since each of the $M$ modes couples inputs and outputs across all frequencies. The total complexity is therefore

$$\mathcal{O}\big((C+K)N \log N \;+\; M(C+K)N\big).$$

For comparison, a standard spatial convolution with kernel size $d \times d$ has cost

$$\mathcal{O}(CKNd^2).$$

SONIC is thus particularly attractive for large receptive fields (where $d$ is large or even global), since the cost remains manageable and the parameter count remains compact.

## 4    EMPIRICAL VALIDATION

**SynthShape.**    We introduce SynthShape, a $64 \times 64$ synthetic segmentation benchmark that measures robustness to rescaling, rotation, translation, distortion, and Gaussian noise under 5-fold cross-validation. We also introduce **HalliGalli**, a spatial-reasoning task requiring long-range dependency modelling: a central patch must be classified according to whether exactly two matching shapes appear in the four distant corners; the centre carries no class signal. Purely local models fail since no local receptive field can capture the relevant structure. SONIC solves HalliGalli and remains robust under noise ($\sigma{=}0.1$), demonstrating that its oriented spectral modes effectively capture global structure without accumulating noise. Implementation details are in Appendix A.1.

Table 1: Comparison of ConvNet, ViT, S4ND, NIFF, GFNet, and SonicNet performance on SynthShape under geometric variations (left), and qualitative examples from SynthShape and HalliGalli-SRT (right).

| Experiment | Value | ConvNet | ViT | S4ND | NIFF | GFNet | SonicNet |
|---|---|---|---|---|---|---|---|
| **Parameter count (M)** | | 0.153 | 0.471 | 0.186 | 0.042 | 1.218 | 0.073 |
| **GMACs** | | 0.156 | 0.012 | 0.023 | 0.041 | 0.139 | 0.006 |
| **Distortion** | 2.0 | 0.95 | 0.83 | 0.84 | 0.95 | 0.55 | 0.95 |
| | 4.0 | 0.94 | 0.81 | 0.83 | 0.95 | 0.54 | 0.94 |
| | 6.0 | 0.93 | 0.81 | 0.82 | 0.94 | 0.53 | 0.94 |
| **Gaussian Noise ($\sigma$)** | 0.1 | 0.98 | 0.72 | 0.89 | 0.99 | 0.67 | 0.97 |
| | 0.2 | 0.92 | 0.44 | 0.67 | 0.96 | 0.37 | 0.90 |
| | 0.3 | 0.76 | 0.31 | 0.43 | 0.86 | 0.21 | 0.71 |
| **Rescaling** | 0.75 | 0.76 | 0.65 | 0.49 | 0.78 | 0.40 | 0.86 |
| | 1.00* | 0.99 | 0.82 | 0.93 | 1.00 | 0.79 | 0.98 |
| | 1.50 | 0.59 | 0.64 | 0.32 | 0.65 | 0.33 | 0.93 |
| **Rotation (°)** | 15 | 0.71 | 0.64 | 0.68 | 0.76 | 0.42 | 0.80 |
| | 30 | 0.28 | 0.33 | 0.50 | 0.30 | 0.29 | 0.24 |
| | 45 | 0.27 | 0.31 | 0.44 | 0.29 | 0.28 | 0.23 |
| **Translation (%)** | 10 | 0.92 | 0.81 | 0.76 | 0.92 | 0.48 | 0.93 |
| | 20 | 0.96 | 0.82 | 0.76 | 0.97 | 0.67 | 0.95 |
| | 30 | 0.92 | 0.80 | 0.77 | 0.92 | 0.52 | 0.92 |
| **Combined** | 10 | 0.80 | 0.74 | 0.47 | 0.81 | 0.35 | 0.87 |
| | 20 | 0.62 | 0.60 | 0.30 | 0.63 | 0.30 | 0.78 |
| | 30 | 0.43 | 0.44 | 0.24 | 0.43 | 0.26 | 0.55 |
| **HalliGalli** | | 0.42 | 0.33 | 0.62 | 1.00 | 0.71 | 1.00 |
| **HalliGalli ($\sigma = 0.1$)** | | 0.33 | 0.33 | 0.49 | 0.56 | 0.37 | 0.86 |

\* Validation accuracy on the training task.

**SynthShape**

**HalliGalli**

**3D medical image segmentation.** Following Isensee et al. (2024), we benchmark on KiTS (kidney/tumour) and ACDC (cardiac) with identical 5-fold splits, preprocessing, augmentations, and postprocessing, so that observed differences are attributable solely to the network backbone. Table 2 shows that SonicNet achieves segmentation quality on par with nnU-Net ResEnc L (DSC 88.55 vs. 88.98 on KiTS; 92.02 vs. 91.40 on ACDC), while using only 2.59 M parameters—an order of magnitude fewer than any competing method. The large parameter gap reflects both SONIC's dimension-independent mode count and the directional regularity of anatomical structures (organ boundaries, vessels), which orientation-aware spectral modes capture efficiently. SonicNet also surpasses all literature Transformer-based architectures (SwinUNETRV2, nnFormer, CoTr) and the Mamba-based U-Mamba Bot in both DSC and NSD, demonstrating that continuous spectral operators transfer effectively to volumetric medical data.

Table 2: **3D medical image segmentation results** (5-fold CV; mean across 5 folds). Columns report **DSC** and **NSD** (at 2 mm). "RT" is runtime (hours) and "VRAM" is peak memory (GB). Literature results are shown in gray as reported by Isensee et al. (2024).

| | KiTS | | ACDC | | | | |
|---|---|---|---|---|---|---|---|
| **Method** | DSC | NSD | DSC | NSD | **Params (M)** | **RT (h)** | **VRAM (GB)** |
| nnU-Net ResEnc L | 88.98 | 85.74 | 91.40 | 96.21 | 31.12 | 34 | 23 |
| **SonicNet (Ours)** | 88.55 | 81.19 | 92.02 | 96.07 | 2.59 | 67 | 61.37 |
| nnU-Net ResEnc L | 88.17 | 85.93 | 91.69 | 95.11 | 31.12 | 36 | 36.60 |
| MedNeXt L k5 | 87.74 | 85.67 | 92.62 | 96.09 | 55.00 | 233 | 18.00 |
| STU-Net L | 85.84 | 83.02 | 89.34 | 95.12 | 440.30 | 51 | 26.50 |
| SwinUNETRV2 | 84.14 | 80.11 | 86.24 | 95.15 | 72.80 | 15 | 13.40 |
| nnFormer | 75.85 | 69.43 | 81.55 | 95.83 | 150.0 | 8 | 5.70 |
| CoTr | 84.59 | 80.92 | 88.02 | 93.74 | 41.27 | 18 | 8.20 |
| U-Mamba Bot | 86.22 | 83.27 | 89.13 | 95.40 | 64.00 | 24 | 12.40 |

**External validation.**  To assess generalisability beyond the training cohort, we evaluate SONIC trained on PI-CAI (Saha et al., 2022) against nnU-Net on two held-out external datasets: Prostate158 (Adams et al., 2022) and PROMIS (Ahmed et al., 2017). As shown in Table 3, SonicNet improves detection across nearly every metric on both datasets while using approximately $12\times$ fewer parameters than nnU-Net. The gains are especially pronounced on the challenging PROMIS cohort, suggesting that SONIC's compact spectral representation generalises more robustly to out-of-distribution data. Qualitative results are in Appendix B.

Table 3: **External validation performance on Prostate158 and PROMIS.** SonicNet achieves improved detection performance with substantially fewer parameters.

| | Metric | nnU-Net | SonicNet |
|---|---|---|---|
| | TRAINABLE PARAMETERS (M/MB) | 31.20/342.0 | 2.59/28.4[*] |
| **Prostate158** | AUROC | 0.814 | **0.841** |
| | AP | 0.533 | **0.548** |
| | F1 Score | 0.632 | **0.649** |
| | Sensitivity | 0.475 | **0.495** |
| | Precision | 0.941 | **0.943** |
| | TP/FP/FN (%) | 0.30/0.02/0.34 | 0.32/0.02/0.32 |
| **PROMIS** | AUROC | 0.646 | **0.687** |
| | AP | 0.195 | **0.258** |
| | F1 Score | 0.185 | **0.223** |
| | Sensitivity | 0.103 | **0.127** |
| | Precision | **0.912** | 0.907 |
| | TP/FP/FN (%) | 0.05/0.01/0.47 | 0.07/0.01/0.47 |

[*] Approximately $12\times$ fewer parameters than nnU-Net.

**ImageNet classification.**  Table 4 evaluates SONIC on ImageNet-1K in two settings. First, as a drop-in replacement for $3\times3$ convolutions in ResNet-50, where it outperforms GFNet and S4ND under identical conditions. Second, as SonicNet—a 4-stage hierarchical backbone designed natively around spectral operators, where SONIC handles spatial mixing and $1\times1$ convolutions handle channel mixing. This avoids bottleneck projections and batch normalisation placement inherited from optimising convolution cost. SonicNet-S achieves 78.8% Top-1 accuracy with only 1.9 GFLOPs—less than half the compute of ResNet-50 (4.1 GFLOPs) while surpassing its accuracy by 2.7 points. The parameter gap relative to medical imaging is narrower because natural images lack consistent directional structure. Crucially, SonicNet exhibits smaller relative degradation under resolution shifts (Figure 3), confirming the resolution invariance from Section 3. Note that DeiT-S cannot be evaluated at non-training resolutions due to its fixed positional embeddings.

Table 4: **ImageNet-1K classification.** Published baselines, ResNet-50 with drop-in spectral operators, and SonicNet variants. All evaluated at $224\times224$.

| | Complexity | | Accuracy (%) | |
|---|---|---|---|---|
| **Model** | Params (M) | GFLOPs | **Top-1** | **Top-5** |
| ResNet-50 | 25.6 | 4.1 | 76.1 | 92.9 |
| DeiT-S/16[†] | 22.1 | 4.6 | 79.9 | 95.0 |
| GFNet-S[†] | 25.0 | 4.5 | 80.0 | 94.9 |
| NIFF[†] | 18.6 | – | 79.7 | 94.8 |
| ResNet-50 + GFNet | 15.7 | 4.6 | 75.1 | 92.2 |
| ResNet-50 + S4ND | 16.7 | 4.6 | 57.5 | 80.3 |
| ResNet-50 + SONIC | 15.2 | 3.5 | 76.7 | 92.9 |
| **SonicNet-T (Ours)** | **5.7** | **0.9** | **75.8** | **92.1** |
| **SonicNet-S (Ours)** | **15.0** | **1.9** | **78.8** | **94.5** |

[†]Published results

Figure 3: Relative performance degradation under resolution changes on ImageNet.

**Learned filter structure.** Figure 4 (left) shows randomly sampled spatial kernels obtained by inverse-transforming the learned spectral filters from each stage of the ImageNet backbone. Early stages learn compact, oriented filters, capturing edges and texture at diverse orientations. With increasing depth, filters progressively broaden: Stage 2 exhibits bipolar lobes sensitive to medium-range structure, Stage 3 develops multi-scale directionally selective patterns, and by Stage 4 filters span nearly the full spatial extent. This hierarchy emerges without supervision on filter shape, confirming that the continuous spectral parameterisation provides an effective inductive bias for multi-scale directional representations. To quantify effective spatial reach we report the normalised 95%-energy radius $R_{95} = r_{95}/(\frac{1}{2}\sqrt{H^2 + W^2})$, where $r_{95}$ is the smallest disk radius $r$ satisfying $\int_{\|\mathbf{x}\| \leq r} |h(\mathbf{x})|^2\, d\mathbf{x} \geq 0.95 \int |h(\mathbf{x})|^2\, d\mathbf{x}$, $h = \mathcal{F}^{-1}[\widehat{H}]$ is the spatial kernel corresponding to the learned spectral filter, and $H \times W$ are the spatial dimensions of the feature map. Thus $R_{95} = 1$ means the kernel's energy spans the full spatial extent (Figure 4, right).

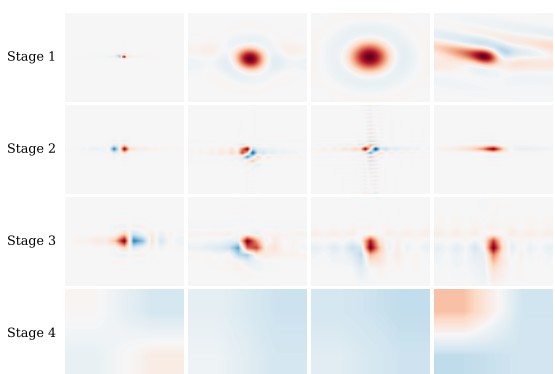 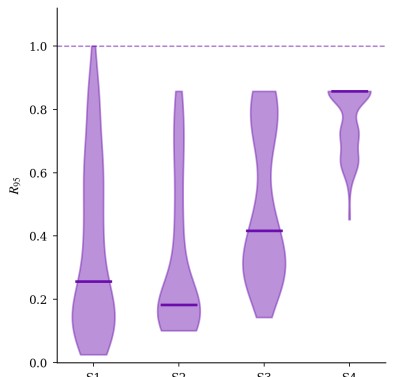

Figure 4: (*Left*) Learned SONIC spatial kernels (4 per stage, ImageNet backbone) via inverse FFT of the spectral filters. Early stages learn compact Gabor-like filters; deeper stages develop broad, globally reaching patterns. (*Right*) Normalised $R_{95}$ reach across all 128 modes per stage. Median reach grows monotonically with depth, approaching full spatial extent (dashed line) by Stage 4.

**Compute and memory overhead.** Figure 5 compares a single SONIC block against a $3\times3$ convolutional block and a ViT self-attention block (patch size $4\times4$, matching the spatial resolution of feature maps). At $224\times224$, SONIC is $1.23\times$ slower and uses $1.18\times$ more memory than convolution while self-attention grows quadratically. At higher resolutions the gap narrows further: by $512\times512$, SONIC is only $1.09\times$ slower than convolution. The overhead remains constant relative to feature map size. Runtime and memory scale linearly in both $C$ and $M$ (Figure 7, Appendix B), making SONIC practical for high-resolution inputs where global context is essential and one does not want to pay the quadratic cost of attention.

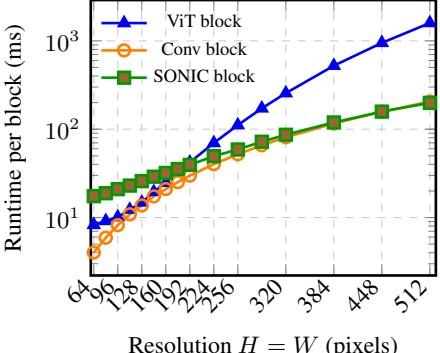 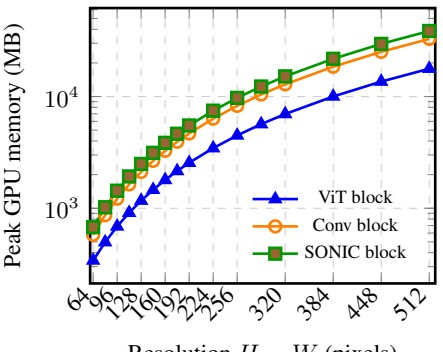

Figure 5: (*Left*) Runtime per block (log scale) and (*Right*) peak GPU memory (log scale) for ViT, convolutional, and SONIC blocks across spatial resolutions. SONIC scales favourably with resolution due to its $\mathcal{O}(N \log N)$ FFT complexity, whereas self-attention grows quadratically.

## 5 DISCUSSION

We introduced SONIC, a spectral factorisation framework that serves as a theoretically grounded, parameter-efficient alternative to spatial convolution blocks. By employing low-rank, orientation-aware operators in the frequency domain, SONIC provides global receptive fields with a principled inductive bias for long-range, structured interactions. Empirically, SONIC demonstrated superior robustness to geometric transformations on SynthShape, solved strict long-range dependencies in HalliGalli within a single block, and matched or exceeded state-of-the-art 3D medical segmentation (KiTS, ACDC) with $< 10\%$ of the parameters of nnU-Net and MedNeXt.

Important limitations remain. Nonlinearities must be applied in the spatial domain, forcing repeated FFT/IFFT operations shared by most spectral architectures. We also observed occasional initialisation instabilities when identical spatial dimensions correspond to different physical scales across datasets, an issue that warrants a more robust initialisation scheme. Finally, the global nature of frequency-domain representations can limit the capture of very fine local structure, suggesting that hybrid spectral-spatial architectures merit further investigation. In summary, SONIC offers a new building block that complements existing paradigms through its long-range receptive field, parameter efficiency, and orientation-awareness, while future work should focus on improving computational efficiency and exploring hybrid designs.

### ACKNOWLEDGMENTS

The authors would like to acknowledge the Research High Performance Computing (RHPC) facility of the Netherlands Cancer Institute (NKI). In this paper, we used large language models to refine wording and improve the clarity of information transfer. All conceptual ideas, discussion of related work, and factual content were developed manually by the authors; the models were employed solely for assistance with presentation.

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

# A IMPLEMENTATION DETAILS

We constrain $s_m > 0$, $\tau_m > 0$, $\mathrm{dc}_m > 0$, and enforce $\mathrm{Re}(a_m) < 0$ so that the spatial response function decays rather than grows. Direction vectors $\boldsymbol{v}_m$ are initialised evenly spaced in $(0, \pi)$ for 2-D, or via a Fibonacci lattice on the sphere for 3-D, with optional jitter.

All parameters are learned end-to-end by backpropagation. A convenient reparameterisation that enforces the constraints is:

$$s_m = \exp(\sigma_m), \quad \tau_m = \exp(\tilde{\tau}_m), \quad a_m^{\mathrm{re}} = -\mathrm{softplus}(\alpha_m), \quad a_m^{\mathrm{im}} = \beta_m,$$

with free variables $\sigma_m, \tilde{\tau}_m, \alpha_m, \beta_m \in \mathbb{R}$. Direction vectors are parameterised via spherical angles $(\theta_m, \phi_m)$ converted to unit vectors. The mixing matrices $B$ and $C$ are complex-valued and learned without constraints.

*Implementation notes.* (i) We standardize each input sample to zero mean and unit variance over all channels and spatial dimensions, with a small noise floor for numerical stability. (ii) We use real-FFT (rFFT/irFFT) along the spatial dimensions; consequently we enforce Hermitian consistency by zeroing the imaginary part at DC and, for even-sized dimensions, at the Nyquist bin. (iii) Direction vectors are rescaled by $D_\Delta^{-1}$ and renormalised to unit length before use, ensuring invariance to pixel spacing. (iv) Optional mode dropout is applied to the filtered modes as a regulariser. (v) Transfer function magnitudes are soft-clamped via $\tanh$ saturation to prevent resonance peaks from destabilising training.

**Butterworth bandlimiting.** To suppress aliasing near the Nyquist limit, we multiply the denominator of $T_m$ by a Butterworth factor $(1 + (\|\boldsymbol{\omega}\|/\omega_{c,m})^{2n})$, where $\omega_{c,m} = b_m \cdot \omega_{\mathrm{nyq}}$ and $b_m = \sigma(\tilde{b}_m) \in (0, 1)$ is a learnable per-mode bandwidth. We use order $n=4$ in all experiments. This allows each mode to adapt how much of the spectrum it utilises.

## A.1 SYNTHSHAPE

The dataset consists of a random number of geometric primitives (circle, square, triangle, cross, star) at random positions and scales within the image, while preventing overlaps through collision checks. Each object is assigned a randomly perturbed base colour, ensuring that models cannot exploit a trivial mapping between RGB values and semantic classes. The ground-truth segmentation mask assigns a unique class label to each shape type, with background indexed as class 0.

**Models.** All models use an embedding width of $c=128$

- **ConvNet:** A lightweight stack of $L$ convolutional layers (default $L = 4$), each followed by group normalisation and GELU activations. A $1 \times 1$ convolution projects the final feature map to the number of classes. The patch size is set to 16 to give the model a fair opportunity to capture broader context, rather than learning solely from small local receptive fields.

- **ViT:** A Vision Transformer consisting of a patch embedding layer, sinusoidal positional encodings (interpolated if image resolution differs), and a stack of transformer blocks with multi-head self-attention and MLP layers. The output features are reshaped and upsampled to the original spatial resolution, followed by a $1 \times 1$ convolution for classification.

- **SonicNet:** For SonicNet we use a depth of 4 stacked SonicBlocks, each consisting of GroupNorm, GELU, and a residual spectral convolutional mapping. The final stage applies GroupNorm, GELU, and a $3\times3$ convolution to project features to class logits.

- **GFNet:** Each block replaces local convolutions by a learned complex-valued mask applied in the Fourier domain. The overall encoder–head structure is kept identical to the ConvNet.

- **NIFF:** Each block learns a continuous frequency response via a small MLP that maps frequency coordinates to complex filter values. These filters are applied depthwise in the Fourier domain and wrapped in the same normalisation, residual, and head structure as the ConvNet.

- **S4ND:** A state-space baseline where the convolutional backbone is replaced by stacked S4ND layers operating directly on the $H \times W$ grid. Each block applies a 2D structured state-space update to the feature map and is embedded in the same residual and segmentation head pattern as the other models.

**Training.** All models were trained using the AdamW optimizer with learning rate $10^{-2}$ and weight decay $10^{-4}$, for 1000 epochs and batch size 32. A one-cycle learning rate schedule was applied. To account for class imbalance, inverse-frequency class weights were computed dynamically from a large synthetic batch and used in the cross-entropy loss. The final training objective combined cross-entropy with the multi-class Dice loss in equal weighting.

**Evaluation.** Model robustness was assessed by applying five geometric transformations at inference: rescaling, rotation, translation, distortion, and Gaussian noise. Each transformation was applied with three levels of severity. Rescaling resized the full image before resampling it back to $64 \times 64$, introducing interpolation artefacts. Translation shifted the input by a fixed percentage of image width/height, potentially moving parts of objects out of frame. Distortion was implemented via bicubic upsampling of a low-resolution displacement field. Rotation was performed around the image centre, and Gaussian noise was added per pixel channel.

## A.2 IMAGENET

We evaluate two settings on ImageNet-1K:

**Drop-in replacement (ResNet-50 + SONIC).** We take a standard ResNet-50 and replace each $3\times3$ convolution in the bottleneck blocks with a SONIC layer, keeping the $1\times1$ projections and skip connections unchanged. This isolates the effect of the spectral operator within an existing convolutional template.

**Native backbone (SonicNet).** We design a 4-stage hierarchical backbone natively around spectral operators. Each stage consists of SonicBlocks: GroupNorm $\to$ SONIC (spatial mixing) $\to$ LayerScale $\to$ residual, followed by an optional $1\times1$ conv MLP for channel mixing. We provide two configurations: SonicNet-T (tiny: depths $[2, 2, 6, 2]$, dims $[64, 128, 256, 512]$, $\sim$5.7M params) and SonicNet-S (small: depths $[3, 3, 9, 3]$, same dims, $\sim$15M params).

Both settings use AdamW with cosine learning rate decay. Full architectural details and the PyTorch implementation are provided in the repository.

## A.3 MEDICAL IMAGING BENCHMARK

**Setup.** Following the recommendations of Isensee et al. (2024), we minimize confounding factors and keep the experiment as plain as possible. We retain the baseline nnU-Net preprocessing and postprocessing and change only the network backbone: the original U-Net is replaced by a stack of SONIC Blocks ("SonicNet"). The first block lifts the input from $C$ to $K$ channels; the remaining $D - 1$ blocks keep $K$ channels. We apply GroupNorm and GELU before a final $3\times3$ convolution to produce $n_{\text{classes}}$ output channels. For this experiment, we used four stacked SonicBlocks (i.e., a depth of 4). We employed stochastic gradient descent with an initial learning rate of $10^{-2}$ and a weight decay of $10^{-5}$. Training was performed with a mini-batch size of two for a total of 1000 epochs, each consisting of 250 iterations. For inference, we used the checkpoint corresponding to the highest validation performance during training.

## A.4 QUALITATIVE COMPARISON OF THE EXTERNAL VALIDATION

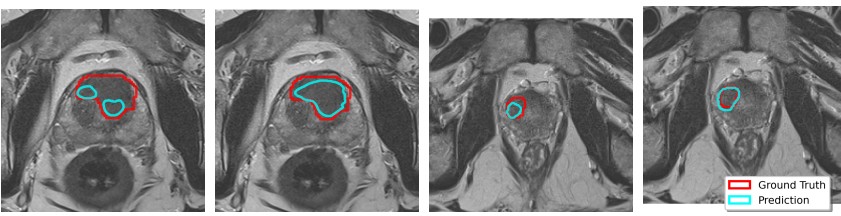

Figure 6: **Qualitative comparison of prostate cancer detection methods.** The figure shows representative cases from the Prostate158 (left) and PROMIS (right) datasets, with ground truth lesions (red) and model predictions (cyan) overlaid on T2-weighted MRI slices (confidence $\geq 0.5$).

## B  PRACTICAL IMPLEMENTATION

**Role of $K$ and $M$.** The parameters $K$ and $M$ play complementary roles in shaping the behaviour of a SONIC block. The number of modes $M$ determines the spectral diversity of the operator; in contrast, the channel width $K$ controls the capacity with which these shared modes are mixed across feature channels. The ratio between $K$ and $M$ therefore reflects the balance between channel-mixing capacity and spectral richness. Understanding this trade-off helps guide architectural choices across different model sizes.

**Practical scalability of SONIC block.** To validate that the SONIC block exhibits the intended linear scaling behavior, we empirically benchmark its runtime and memory usage across a range of channel dimensions and mode counts. We confirm this behavior by measuring wall-clock runtime and peak memory under controlled synthetic settings, sweeping C and M independently while keeping spatial resolution fixed. Across all tested configurations, runtime increases as a straight line with respect to both variables, and memory usage follows the same linear trend.

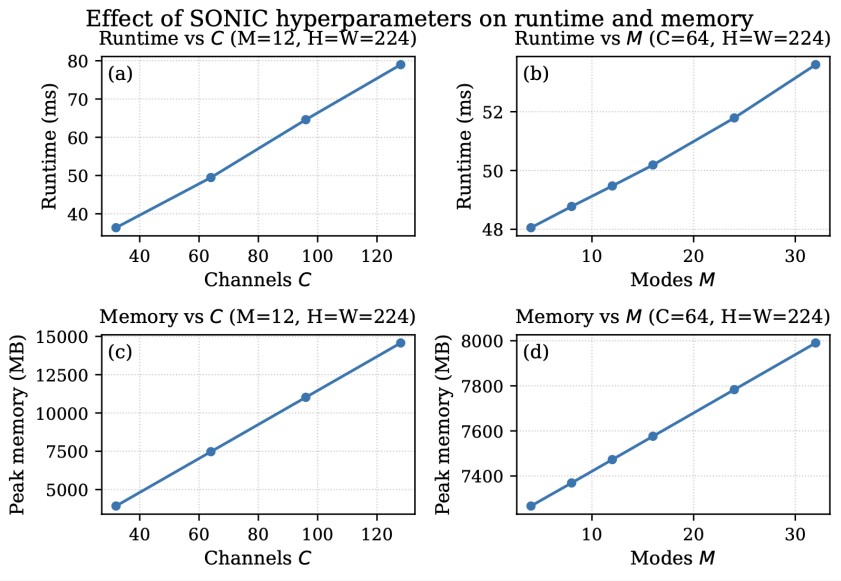

Figure 7: Runtime and memory of SONIC when varying channels $C$ and modes $M$ at fixed resolution.

## C  SUPPORTING PROOFS

### C.1  CONVOLUTION THEOREM FOR THE $D$-DIMENSIONAL FOURIER TRANSFORM

Let the convolution of two functions on $\mathbb{R}^D$ be defined by

$$(f * g)(\mathbf{x}) := \int_{\mathbb{R}^D} f(\boldsymbol{\tau})\, g(\mathbf{x} - \boldsymbol{\tau})\, d\boldsymbol{\tau}, \qquad \mathbf{x} \in \mathbb{R}^D.$$

Then, for $\boldsymbol{\omega} \in \mathbb{R}^D$, the $D$-dimensional Fourier transform satisfies

$$\mathcal{F}\{f * g\}(\boldsymbol{\omega}) = \int_{\mathbb{R}^D} \left( \int_{\mathbb{R}^D} f(\boldsymbol{\tau})\, g(\mathbf{x} - \boldsymbol{\tau})\, d\boldsymbol{\tau} \right) e^{-i\boldsymbol{\omega}\cdot\mathbf{x}}\, d\mathbf{x}$$

$$= \int_{\mathbb{R}^D} \int_{\mathbb{R}^D} f(\boldsymbol{\tau})\, g(\mathbf{u})\, e^{-i\boldsymbol{\omega}\cdot(\boldsymbol{\tau}+\mathbf{u})}\, d\mathbf{u}\, d\boldsymbol{\tau}$$

$$= \left( \int_{\mathbb{R}^D} f(\boldsymbol{\tau})\, e^{-i\boldsymbol{\omega}\cdot\boldsymbol{\tau}}\, d\boldsymbol{\tau} \right) \left( \int_{\mathbb{R}^D} g(\mathbf{u})\, e^{-i\boldsymbol{\omega}\cdot\mathbf{u}}\, d\mathbf{u} \right).$$

Hence,

$$\mathcal{F}\{f * g\}(\boldsymbol{\omega}) = \mathcal{F}\{f\}(\boldsymbol{\omega})\,\mathcal{F}\{g\}(\boldsymbol{\omega}).$$

## C.2 CONNECTION TO STATE-SPACE KERNELS

Consider the linear time-invariant state-space model

$$\dot{x}(t) = Ax(t) + Bu(t), \qquad y(t) = Cx(t), \tag{17}$$

with $x(t) \in \mathbb{C}^n$, $u(t) \in \mathbb{C}^m$, $y(t) \in \mathbb{C}^p$, and system matrices $A \in \mathbb{C}^{n \times n}$, $B \in \mathbb{C}^{n \times m}$, $C \in \mathbb{C}^{p \times n}$. Assume a zero initial state $x(0^-) = 0$ and a strictly proper output.

The corresponding impulse response (or kernel) is

$$\mathcal{K}(t) = Ce^{At}B, \qquad t \geq 0. \tag{18}$$

By definition, the transfer function is the Laplace transform of the impulse response:

$$H(s) = \int_0^\infty e^{-st}\,\mathcal{K}(t)\,dt = \int_0^\infty e^{-st}\,Ce^{At}B\,dt. \tag{19}$$

Pulling out $C$ and $B$ gives

$$H(s) = C\left(\int_0^\infty e^{(A-sI)t}\,dt\right)B. \tag{20}$$

For $\mathrm{Re}(s)$ sufficiently large, the integral converges to

$$\int_0^\infty e^{(A-sI)t}\,dt = (sI - A)^{-1}. \tag{21}$$

Hence the transfer function is

$$H(s) = C(sI - A)^{-1}B. \tag{22}$$

**SONIC with Restricted Modes.** We show that our general Fourier domain formulation reduces to the Laplace resolvent parameterisation of S4ND when orientations are restricted to the coordinate axes.

Recall our frequency response factorisation

$$\widehat{H}_{c,k}(\boldsymbol{\omega}) = \sum_{m=1}^M C_{km}\,T_m(\boldsymbol{\omega})\,B_{mc}, \tag{23}$$

with mode response

$$T_m(\boldsymbol{\omega}) = \frac{1}{is_m(\boldsymbol{\omega}\cdot\boldsymbol{v}_m) \;-\; a_m \;+\; \tau_m\|(I - \boldsymbol{v}_m\boldsymbol{v}_m^\top)\boldsymbol{\omega}\|^2}. \tag{24}$$

Suppose $\boldsymbol{v}_m = e_d$, the $d$-th standard basis vector. Then

$$\boldsymbol{\omega}\cdot\boldsymbol{v}_m = \omega_d, \qquad (I - \boldsymbol{v}_m\boldsymbol{v}_m^\top)\boldsymbol{\omega} = \sum_{j\neq d}\omega_j e_j,$$

so that

$$\|(I - \boldsymbol{v}_m\boldsymbol{v}_m^\top)\boldsymbol{\omega}\|^2 = \sum_{j\neq d}\omega_j^2.$$

In this case,

$$T_m(\boldsymbol{\omega}) = \frac{1}{is_m\,\omega_d - a_m + \tau_m\sum_{j\neq d}\omega_j^2}. \tag{25}$$

We discard the transverse penalty $\tau_m = 0$, then

$$T_m(\omega_d) = \frac{1}{is_m\omega_d - a_m} = \frac{1}{s_m}\,\frac{1}{i\omega_d - \frac{a_m}{s_m}},$$

where the absorption is into the learned parameters ($a_m/s_m$ in $A$, and $B$ or $C$ absorb $1/s_m$). Thus

$$\widehat{H}_{c,k}(\omega_d) = \left[C\,(i\omega_d I - A)^{-1}B\right]_{kc}, \qquad H(s) = C(sI - A)^{-1}B.$$

