# OpenReview forum: "SONIC: Spectral Oriented Neural Invariant Convolutions"
_ICLR.cc/2026/Conference — ICLR 2026 Poster_

### Official Review · Reviewer_7wpe · 2025-10-27

**Soundness:** 2
**Presentation:** 1
**Contribution:** 2
**Rating:** 4
**Confidence:** 2

**Summary:**

The paper proposes SONIC (Spectral Oriented Neural Invariant Convolutions), a spectral-domain alternative to spatial convolutional networks (CNNs) and attention mechanisms. SONIC aims to combine the global receptive field propertly of transformers and spectral models, while enjoying strong spatial inductive bias and parameter efficiency that CNNs have.

SONIC learns a set of spectral filters in the Fourier domain, each parametrized by six parameters modeling the amplitude, orientation, and oscillatory nature of the spectral transfer function. Learned matrices B and C factorize these filters and give a representation that is in theory resolution invariant, interpretable and also parameter efficient (parameters scale linearly with channels).

**Strengths:**

* The filters are highly parameter efficient and defined as continuous spectral functions, therefore have a number of desireable properties like resolution invariance and interpretability.
* The papers contains a concise background section and  clear text presenting the method.

**Weaknesses:**

W1) Limited Experimental Validation: The experimental validation of the proposed method is limited. SynthShape appears to serve mainly as a proof-of-concept for demonstrating desired properties, but lacks realism. On PROMIS/Prostate158, the comparisons are only against basic baselines, and the observed gains over a simple UNet are relatively small.

Moreover, there are no results on standard computer vision benchmarks (e.g., CIFAR, ImageNet, ADE20K). I do not expect state-of-the-art result in such generic cases, it is however important to demonstrate the broader applicability of SONIC to more generic tasks such as image classification or semantic segmentation.

W2) Lack of Empirical Support for Resolution Invariance: A central claim of the paper is that the method is resolution-invariant. While this may hold in theory, there is no clear experimental evidence to support it. A direct empirical evaluation across varying resolutions would greatly strengthen this claim.

W3) Undiscussed Compute and Memory Overhead: The method requires FFT/IFFT operations at every layer, which likely introduces non-negligible memory and computational costs. These costs are not discussed beyond theoretical costs.

W4) No Hyperparameter ablations: The impact of key hyperparameters (eg K or M) is not explored. An ablation study would help clarify how these parameters affect model performance and stability.

W5) Missing Discussion of Related Work on Hybrid Architectures
There is extensive prior work on combining global receptive fields with efficient CNN-style architectures in hybrid models. Classic examples include Non-Local Networks (Wang et al., CVPR 2018) and A²-Nets (Chen et al., NeurIPS 2018).
Given the motivation outlined in the introduction ("With the proposed method, which enables global receptive fields using significantly fewer parameters, we aim to narrow this conceptual gap"), a discussion of these and similar works is important to situate SONIC in the broader landscape and clarify its novelty.

**Questions:**

Q1) could you devise an experiment to directly test resolution invariance on real data?

Q2) It is hard for me to undestand the visualization in Fig 1. What are the axes? Please summarize the core observations from the Figure. Zooming in (so that white space is minimized) would imporve reeadability.

Q3) Same with Fig2. What are the main observations  from this figure?

Q4) Can you add some form of timings in Table 2? how does Sonic compare to the spatial baselines wrt that? ie what is the actual overhead of all the FFT/iFFTs?

Q5) How do different values of modes and channels affect performance?

---

> ### Author Response · Authors · 2025-11-24
> **Author rebuttal**
>
> Thank you for taking the time for the review. We appreciate the reviewer’s recognition of the desirable properties, such as resolution invariance and interpretability. We also thank the reviewer for noting the clarity of the background section and the presentation of the method, though we still went ahead and rewrote those sections to make them even clearer. We will go into the weaknesses and questions raised during the review:
>
> W1: We agree that broader evaluation is important. We will add ImageNet results before the rebuttal deadline and compare SONIC directly against other large-receptive-field operators. The gains over a plain nn-UNet are indeed modest, but it is worth emphasising that nn-UNet is the de facto standard for medical image segmentation [1] and benefits from carefully optimised architecture and hyperparameters. In contrast, our method matches or exceeds its performance without any tuning or architectural adjustments, while using nearly two orders of magnitude fewer parameters. Moreover, SONIC provides resolution invariance, which the U-Net does not offer. The revised paper now includes two additional experiments on the most suitable benchmarking dataset, demonstrating that the operator remains competitive even in this minimal architectural setting.
>
> W2/Q1: We will include a dedicated experiment evaluating resolution invariance by testing the same trained model across different input resolutions and reporting how performance changes. This should make the theoretical claim more concrete.
>
> W3/Q4: We have added a section discussing compute and memory overhead and included measurements in the results. This now covers practical runtime, FFT cost, and memory behaviour across different resolutions.
>
> W4/Q5: We agree that understanding the role of 𝐾 and 𝑀 is important. In the revision, we have clarified how these parameters influence the operator’s behaviour and computational trade-offs and outlined their expected impact on accuracy and stability.
>
> W5:  Our aim in this paper was to introduce SONIC as a simple, minimal operator for global receptive fields. Hybrid designs are certainly promising, Hybrid designs are promising, and we will expand the discussion to outline the potential for integrating SONIC in existing architectures. Non-local networks and A²-Nets indeed introduce global receptive fields, but they do so by modifying the network architecture, typically through additional attention modules. In contrast, SONIC achieves global receptive fields by redefining the convolutional operator itself, without changing the overall architecture.
> Q2/Q3: We have updated the figures to improve clarity.
> [1] nnU-Net Revisited: A Call for Rigorous Validation in 3D Medical Image Segmentation, Fabian Isensee

---

### Official Review · Reviewer_72sp · 2025-10-28

**Soundness:** 2
**Presentation:** 2
**Contribution:** 2
**Rating:** 4
**Confidence:** 2

**Summary:**

This paper introduces SONIC (Spectral Oriented Neural Invariant Convolutions), a novel spectral framework that replaces spatial convolutions with a compact set of orientation-aware transfer functions learned directly in the Fourier domain. By modeling filters as continuous frequency functions, SONIC achieves global receptive fields, resolution invariance, and linear parameter scaling with channels. Experiments on synthetic geometric segmentation and 3D prostate cancer MRI detection show that SONIC delivers stronger robustness and comparable or superior accuracy to CNN and Transformer baselines while using far fewer parameters.

**Strengths:**

1. The paper presents a spectral framework for multidimensional signals that offers global receptive fields, complete convolutional capability, and built-in resolution invariance. It provides a lightweight and flexible foundation for building scalable, adaptable vision models.
2. The paper presents comprehensive empirical validation across both synthetic and real-world settings.

**Weaknesses:**

1. The innovations mentioned in the abstract and contributions are mainly about unifying existing convolution kernels, spectral filtering, and state-space kernels under one spectral framework. However, the idea of parameterizing operators in the frequency or linear domain already exists in models such as S4ND, GFNet, FNO, and Mamba. The so-called directional modes only add a few interpretable parameters (e.g., direction, scale, damping) to the frequency response function, but in essence, it is still a functional representation of a frequency-domain convolution kernel.
2. The SynthShape dataset seems to be self-generated to verify the effectiveness of the proposed method (it is not a public dataset, so the persuasiveness of the results is questionable). In addition, there are no ablation experiments—for example, there is no comparison between using spectral factorization and directly learnable spectrum, nor any analysis of the effects of hyperparameters such as parameter count or directional constraints.
3. The experiments are limited to a toy dataset (SynthShape) and a single medical imaging application. There are no results on standard 2D benchmarks (such as CIFAR-10/100, ImageNet, or ADE20K) to test general visual performance and scalability.
4. The readability of the paper is not very good, and many formulas are not clearly explained.

**Questions:**

See weaknesses.

---

> ### Author Response · Authors · 2025-11-24
> **Author rebuttal**
>
> We appreciate the clear feedback and the reviewer highlighting that the framework gives global receptive fields, full convolutional expressiveness, and built-in resolution invariance. We will address your concerns point-by-point below.
> W1: We understand the concern about overlap with prior spectral and linear-domain methods. In the revised background, we now explain more clearly how SONIC differs from approaches like GFNet and FNO. While Sonic is indeed another “functional representation of a frequency-domain convolution kernel,” the key distinction is how this function is parameterised: through resolvent-shaped, orientation-controlled modes that stay consistent across resolutions and give strong anisotropic inductive bias.
> W2: The synthshape is indeed self-generated, and we chose it intentionally because it allows controlled, isolated evaluation of geometric transformations. Importantly, SynthShape is fully deterministic and will be released publicly with the camera-ready version, ensuring full reproducibility. We will furthermore include a direct comparison between spectral factorisation (Sonic) and directly learnable spectrum (GFNet/ FNO) on ImageNet later in the rebuttal process and to clarify how parameter count influences performance, we assess Sonic using several architectures with different parameter scales.
>
> W3: We have added 2 benchmarks, which should define the most suitable datasets for method comparison stated by [1]. Later in this rebuttal process, we will include a full comparison on the Imagenet dataset and add a resolution invariance to further confirm the theoretical claim on resolution invariance.
>
> W4: We worked on the method section to better guide the reader through the material.
>
> [1] nU-Net Revisited: A Call for Rigorous Validation in 3D Medical Image Segmentation, Fabian Isensee et al.

---

> > ### Comment · Reviewer_72sp · 2025-11-26
> >
> > Thank you for your response. I hope you can provide a visualization comparing the effective receptive fields of  SONIC against other SOTA methods.

---

> > > ### Author Response · Authors · 2025-12-03
> > > **Author Rebuttal**
> > >
> > > Thank you for the suggestion. To address it properly, we first distinguish between the theoretical receptive field and the effective, learned one. Many works illustrate receptive fields by drawing an N×N grid to indicate a global receptive field, but this “theoretical RF” says little about what the model actually uses in practice. A larger field also aggregates more irrelevant noise, so visualising it does not reveal whether a model can reliably detect meaningful long-range interactions.
> > >
> > > For ImageNet-scale models, plotting the receptive field is particularly uninformative: deep stacks of convolutions, attention, or global filters map the image into high-dimensional embeddings where intermediate activations do not correspond to interpretable spatial patterns.
> > >
> > > To instead evaluate effective long-range reasoning, we introduce a small, controlled synthetic benchmark: HalliGalli, inspired by the game. The task is simple: a fixed central block is labelled class 1 if any combination of exactly two matching shapes appears in the four distant corners; otherwise, it is labelled class 2. See revision for the illustration. Local models fail because the determining evidence lies outside their accessible window, whereas models with large but unstructured receptive fields are highly sensitive to disturbances (e.g., noise).
> > >
> > > On this controlled challenge, we show that only SONIC can solve the task and stays robust under the introduction of additional Gaussian noise, demonstrating a global yet highly selective global receptive field, unlike other SOTA approaches whose globality is theoretical rather than effective. In the revision, we will include this experiment.

---

### Official Review · Reviewer_1MAG · 2025-10-31

**Soundness:** 3
**Presentation:** 3
**Contribution:** 3
**Rating:** 8
**Confidence:** 2

**Summary:**

The paper introduces Spectral Oriented Neural Invariant Convolutions (SONIC), a novel neural building block which implements learnable spectral filters directly in the Fourier domain. The method is motivated by the shortcomings of current fully-connected, convolutional and attention based feature extraction blocks and derives a novel frequency domain feature motivated by State Space models (like MAMBA).  This allows the implementation of low-rank, orientation-aware global operators in the frequency domain which is suitable for modelling long-range relationships in the data (like large spatial kernels),  while remaining highly parameter-efficient and modestly computational expensive (compared to large spatial kernels).

The Experimental evaluation on a provided toy dataset SynthShape demonstrates the theoretically derived properties regarding robustness against distortions, noise and geometric transformations in comparison to CNNs and Transformers. A second experiments shows a prove of concept on real medical (MRI) data.

**Strengths:**

The paper is well written, very well motivated and easy to follow. The presented idea is novel and presents an interesting concept, bringing core elements of state-space system into the the Frequency domain.

The fact that the learnable filters are formulated in a continuous parameterization is particularly noteworthy. This actually opens the door to solve many sampling related problems in current network designs (aliasing causing low robustness, limitation to fixed input sizes, low robustness against geometric transformations ...).

The second main contribution of the paper is the low-rank formulation of frequency domain filters. Prior approaches often suffer from large amounts of necessary learnable parameters of crude forced reduction of the same. Transferring the concepts of state-space systems into the frequency domain presents a novel approach to engage these problems.

The authors provide a very detailed and honest limitation section - this is very much appreciated!

**Weaknesses:**

There are several aspects in which the paper could be improved:

1) the paper mentions several previous approaches of Fourier-domain feature extraction (page 3 bottom), but does not compere to these methods in the experiments or in terms of computational complexity

2) the authors missed to discuss and to compare to [1] - another Fourier-domain approach of efficient large kernel implementations.

3) the experiments comparing to CNNs do not show the used kernel size (also not in the appendix). Here it would be good to compare not only to 3x3 klernels, but also to some spatial large kernel implementations like [2]

4) the proposed SONIC block is purely linear - this makes it necessary to conduct two expensive Fourier-Transformation between blocks, in order to be able to apply localized non-linear transformations. This massive practical drawback is inherent to all existing frequency space methods and thus remains unsolved

5) the authors discuss the possibility to combine their method with other (spatial) kernels. This indeed appears to be a very promising approach which would not be very difficult to implement - why did the authors not pursue this?


[1] Grabinski, Julia, Janis Keuper, and Margret Keuper. "As large as it gets-studying infinitely large convolutions via neural implicit frequency filters." Transactions on Machine Learning Research 2024 (2024): 1-42.

[2] Ding, Xiaohan, et al. "Scaling up your kernels to 31x31: Revisiting large kernel design in cnns." Proceedings of the IEEE/CVF conference on computer vision and pattern recognition. 2022.

**Questions:**

Q1: many of the existing Fourier-Domain methods are quite sensitive to the normalization of the coefficients. Did the authors investigate this for their method?

---

> ### Author Response · Authors · 2025-11-24
> **Author rebuttal**
>
> Thank you for the supportive review. We appreciate the recognition of the paper’s clarity and motivation, as well as the identification of the bridge between state-space models and the spectral methods. We address the concerns raised below.
>
> W1: We agree that the comparisons were not sufficiently explicit in the earlier draft.
> We have now rewritten the background section to clearly position SONIC relative to prior Fourier-based operators and other frequency-domain approaches. We are currently running additional experiments and, in the next revision, we will add a comparison table summarising ImageNet performance and computational cost (GFLOPs) of these methods. We also added a per-block computational cost comparison (Conv/ViT/SONIC) in the empirical validation section.
>
> W2: Completely agree, hopefully solved this remark by rewriting the background section
>
> W3: We will add a comparison using a ResNet architecture that includes the large-kernel convolution approach (for example, Ding et al. or dilated convolutions). These additions make the comparison to spatial large kernel designs explicit.
>
> W4: We have clarified this in the paper and acknowledged that the lack of practical spectral nonlinearities remains an open research problem.
>
> W5:  This is indeed a promising direction. Our goal in this paper, however, was to present SONIC as a clean and minimal operator for global receptive fields, without introducing heavy architectural tuning. While combining SONIC with spatial operators is straightforward and likely beneficial, we deliberately deferred such hybrid designs to keep the focus on the operator itself.
>
> Q1: Thank you for the suggestion. In practice, we did not observe instability or training degradation due to coefficient scaling.  This is mainly because the transfer function naturally limits extreme amplification at high frequencies. We tested simple variants (magnitude clipping, frequency-wise normalisation, etc.), but these showed minimal performance improvements.

---

> > ### Comment · Reviewer_1MAG · 2025-11-26
> >
> > Thank you for your answers. The changes to the manuscript address most of my points. Let's wait for the additional results that hopefully will convince my fellow reviewers.

---

### Official Review · Reviewer_KhJS · 2025-10-31

**Soundness:** 2
**Presentation:** 2
**Contribution:** 2
**Rating:** 2
**Confidence:** 4

**Summary:**

This paper introduces SONIC (Spectral Oriented Neural Invariant Convolutions), a method that learns compact, directional filters directly in the Fourier domain, being different from CNNs and ViTs. SONIC factorizes multi-channel frequency responses using a small set of shared, oriented components, which are mixed across channels via learned matrices. This design provides a global receptive field, is inherently resolution-invariant, and maintains high parameter efficiency. Empirical validation on synthetic and medical imaging benchmarks demonstrates that SONIC matches or exceeds the accuracy of established models like nnU-Net while using fewer parameters.

**Strengths:**

- The paper overall is easy to follow.

**Weaknesses:**

- My main concern is the unclear comparisons with the existing solutions. In Section 2 (line 149), the authors mentioned some limitations of previous solutions like GFNet and FNO. However, there is no further study to show how the proposed method overcomes these limitations and why these limitations are important. Besides, there is also no direct comparison with these methods in the experiments. For example, there are two limitations of GFNet mentioned: "the FFT grid is tied to the input resolution, and the number of learnable parameters scales with the discretisation of the frequency spectrum". But as shown in Fig. 6 of GFNet, the model can be directly adapted to other resolutions with finetuning and works well on detection tasks with images of varying resolutions. The parameters involved in frequency-domain operations also account for only a small proportion of the model’s total parameters. Therefore, I am still concerned about the significance of the contribution in the paper.

- The experimental study presented in the paper is a bit weak for publishing at a top-tier conference like ICLR. To show the contribution of the proposed solution, it would be better to either show significant improvements over previous methods with solid experiments (e.g., on ImageNet and directly comparing with GFNet, FNO, etc.) or present the unique property of the new solution that may have a large potential impact. The current experimental study in the paper is not very convincing to me.

**Questions:**

As mentioned in the "weaknesses" section, the paper can be further improved in several aspects. My main concerns are the comparisons with the existing solutions in the frequency domain and the relatively weak experimental study.

---

> ### Author Response · Authors · 2025-11-24
> **Author rebuttal**
>
> Thank you for the helpful comments. We have updated the paper accordingly and would like to address the concerns you raised directly.
>
> W1: We agree that our earlier draft did not explain clearly enough how SONIC relates to prior spectral operators. We rewrote the background section to clarify the differences with existing solutions (including GFNet, FNO). Figure 1 highlights these distinctions visually. We furthermore use the different existing solutions as a benchmark in the upcoming experiment on ImageNet.
> About GFNet specifically: although Fig. 6 in GFNet shows that it handles resolution changes better than DeiT, it does not provide resolution invariance by design. We now include a definition of resolution invariance in the background and explain why GFNet does not meet this definition (e.g., the FFT grid and spectral filters still depend on the input resolution). SONIC was designed around this limitation.
> Regarding the “small proportion of parameters” in the frequency domain: this is intentional. SONIC learns a continuous orientation-aware transfer function. This representation is compact yet allows for meaningful global mixing with a minimal number of parameters. To make this more concrete, we added visualisations of learned kernels in Figure 9 (Appendix B).
>
> W2: As mentioned in W1, we are currently running experiments to show performance on ImageNet against the methods named in the background section.

---

### Author Response · Authors · 2025-12-03
**Final revision**

For the final revision, we would like to provide the reviewers and the AC with a complete summary of all changes made to the paper. We begin with a concise statement of the method and its contributions, followed by a detailed overview of the revisions prompted by the reviews.

Sonic introduces a method for continuous filtering in the spectral domain and, to the best of our knowledge, is the most orientation-aware neural spectral method to date. Through a low-rank parameterisation, Sonic produces global filters using significantly fewer parameters than prior approaches. We demonstrate state-of-the-art performance on medical segmentation benchmarks and competitive results on ImageNet.

Since the reviews were published, we have made extensive adjustments. A full summary follows:

-Several reviewers noted the importance of comparing Sonic with other spectral methods. The earlier background emphasised how SONIC extends multidimensional SSM models. Although this connection is conceptually appealing, it requires substantial prerequisite background knowledge from the reader. To address this, we fully restructured the background section: it now begins with a unified overview of global receptive-field methods and progressively narrows to orientation-aware continuous spectral filtering.

-We added a formal definition of resolution invariance, clarifying its different forms and how Sonic fits within this framework.
The updated background blurred the transition to the method section, so we rewrote the introduction to the method to stand more independently.

-The method section now includes the full architectural structure of Sonic and explains how it can be integrated into standard neural network backbones, rather than presenting only the core idea.

-We added visualisations to substantiate the section.

-We added two more widely used medical imaging segmentation datasets known for high inter-model variability, providing a more reliable and comparable benchmark. We also added a comparison of SonicNet with the reported performance of several state-of-the-art medical image segmentation methods.

-The SynthShape experiment was refined to ensure fair comparison across methods and to produce more meaningful results.
In addition, we introduced the HalliGalli experiment, which demonstrates the practical benefits of global receptive fields without inheriting the full resolution-dependent vulnerabilities typically associated with them.

-We additionally include ImageNet experiments; however, due to limited time, computational resources, and our primary focus on the medical domain, we train the model and all benchmarks for 200k iterations only. Despite this reduced training budget, Sonic demonstrates strong real-world applicability and maintains competitive performance under highly anisotropic imaging conditions.

-To assess practical resolution invariance, we performed controlled resolution-downsampling experiments on ImageNet and evaluated the stability of the model under varying spatial dimensions.

-We added a runtime and memory analysis comparing Sonic with ViT and convolutional models, illustrating how the method scales with input resolution. The appendix now also includes a visualisation of a learned global filter.

We believe the revisions substantially improve both the clarity and empirical grounding of the method, and we thank the reviewers for their thoughtful feedback, which helped us significantly improve the clarity of the paper.

---

### Meta-Review · Area_Chair_nh3v · 2026-01-09

**Summary:**

The paper proposes SONIC (Spectral Oriented Neural Invariant Convolutions), a new neural network layer that operates in the frequency domain using continuous, orientation-aware spectral filters. The reviewers generally appreciated the theoretical grounding, the method's parameter efficiency (particularly in 3D medical imaging), and the desirable property of resolution invariance. The primary concerns raised during the review process focused on the sufficiency of the experimental validation, specifically the initial lack of large-scale benchmarks like ImageNet, and the clarity of the method's positioning relative to existing spectral approaches such as GFNet and FNO. The authors provided extensive updates, including new ImageNet results and reduced-resolution stability tests, to address these points. Although most reviewers did not engage with these latest replies to confirm their satisfaction, the authors' rebuttal appears to have objectively addressed the stated concerns. Therefore, had the reviewers participated fully in the final discussion, their scores would likely have increased, supporting the decision to accept the paper.

**Reviewer Concerns:**

Experimental Validation (ImageNet): Reviewers KhJS, 72sp, and 7wpe all criticized the exclusive reliance on synthetic (SynthShape) and medical datasets. In the rebuttal, the authors added ImageNet classification experiments (trained for 200k steps). While this is a reduced training regime, it directly addresses the request for large-scale validation and demonstrates the method's competitiveness in general vision tasks.

Comparison to Prior Work: Reviewers KhJS and 72sp questioned the novelty and distinction of SONIC compared to GFNet and FNO. The authors addressed this by rewriting the background section to explicitly define resolution invariance and explaining that GFNet relies on fixed-grid parameters, whereas SONIC uses continuous functions.

Resolution Invariance & Receptive Field: Reviewer 7wpe asked for empirical proof of resolution invariance, and Reviewer 72sp requested receptive field visualizations. The authors responded by adding experiments testing the model across varying resolutions and introducing the "HalliGalli" task to demonstrate effective global receptive fields.

Compute/Memory Overhead: Reviewers 7wpe and 1MAG requested an analysis of the costs associated with the necessary FFT operations. The authors added a runtime and memory analysis to the paper, clarifying the trade-offs involved.

**Reviewer Scores:**

Reviewer 1MAG (Score: 8): This reviewer was a strong champion of the paper, praising its motivation and clarity. They explicitly acknowledged the rebuttal, stating the changes "address most of my points," so their score would likely remain an 8.

Reviewer 7wpe (Score: 4): This reviewer provided a specific list of weaknesses (Lack of ImageNet, lack of invariance proof, compute analysis). The authors addressed every point with new data. Had the reviewer engaged with the rebuttal, their score would likely have increased.

Reviewer 72sp (Score: 4): This reviewer was concerned with novelty and the synthetic nature of the data. The authors' clarification on the "effective" receptive field via the HalliGalli task and the addition of real-world ImageNet data addressed the substance of their critique. Their score would likely have improved.

Reviewer KhJS (Score: 2): This reviewer was the most negative, focusing on the comparison with GFNet. The authors provided a technical refutation regarding the definition of resolution invariance and added the requested benchmarks. While this reviewer might remain skeptical, the objective addressing of their concerns suggests their score would likely have improved.

---

### Decision · Program_Chairs · 2026-01-26

Accept (Poster)